# Effect of Radiofrequency Plasma Spheroidization Treatment on the Laser Directed Energy Deposited Properties of Low-Cost Hydrogenated-Dehydrogenated Titanium Powder

**DOI:** 10.3390/ma15041548

**Published:** 2022-02-18

**Authors:** Min Liu, Liufei Huang, Congcong Ren, Dou Wang, Qiang Li, Jinfeng Li

**Affiliations:** 1Xinjiang Key Laboratory of Solid State Physics and Devices, Xinjiang University, Urumqi 830046, China; liuminchn@foxmail.com; 2School of Physics and Technology, Xinjiang University, Urumqi 830046, China; 3Institute of Materials, China Academy of Engineering Physics, Mianyang 621908, China; lyyhuangliufei@126.com (L.H.); rc13572802927@163.com (C.R.); wangd0005@126.com (D.W.); 4School of Mechanical Engineering, Xinjiang University, Urumqi 830047, China

**Keywords:** hydrogenated–dehydrogenated titanium powder, laser directed energy deposition, microstructure, mechanical property, radio frequency plasma technique

## Abstract

Titanium for additive manufacturing presents a challenge in the control of costs in the fabrication of products with expanding applications compared with cast titanium. In this study, hydrogenated–dehydrogenated (HDH) titanium powder with a low cost was employed to produce spherical Ti powder using the radiofrequency plasma (RF) technique. The spherical Ti powder was used as the raw material for laser directed energy deposition (LDED) to produce commercially pure titanium (CP-Ti). Microstructural analyses of the powder revealed that RF treatment, not only optimized the shape of the titanium powder, but also benefited in the removal of the residual hydride phase of the powder. Furthermore, the LDED-HDH-RF-produced samples showed an excellent combination of tensile strength and tensile ductility compared to the cast and the LDED-HDH-produced samples. Such an enhancement in the mechanical properties was attributed to the refinement of the α grain size and the dense microstructure. The present work provides an approach for LDED-produced CP-Ti to address the economic and mechanical properties of the materials, while also providing insights into the expanding application of HDH titanium powder.

## 1. Introduction

Titanium and its alloys, as engineering materials, are employed in the aerospace, military, and medical fields due to their high specific strength, high specific modulus, and corrosion resistance [1]. Commercially pure titanium (CP-Ti) has better biocompatibility, weldability, and osseointegration than other titanium alloys, which makes it extremely useful [2]. However, it is difficult to melt and process owing to CP-Ti’s notoriously active chemical activity, which limits its large-scale applicability [3].

Components with complex internal structures can be directly produced using additive manufacturing (AM) technology, which provides a powerful tool in the application of titanium and its alloys. Recently, additive manufacturing (AM) fabrication of CP-Ti has received considerable research attention [4]. Many researchers have suggested that, among the large number of metallic materials, CP-Ti is well suited to AM due to AM-produced titanium materials usually presenting attractive mechanical properties, which enables the precision forming of large-sized complex components designed for specific uses [5]. Santos et al. found that adjusting the power of a laser beam and the laser scanning hatch spacing in the AM process helps to improve the density of CP-Ti samples, from ~92% to ~98%, which is beneficial to improving the torsional fatigue strength [6]. Hasib et al. found that that AM-produced CP-Ti samples generally yielded a higher tensile strength and fatigue crack growth resistance compared to wrought material [7,8]. AM technology provides a simple tool to obtain titanium components that are expected to have broad application in medical or chemical-plant industries [9,10].

Previous studies on CP-Ti using AM have focused on parameter optimization to achieve high-density parts and to evaluate their mechanical properties [11]. However, there are limitations to producing corresponding AM-produced titanium components of economical utility for practical applications as the current preparation of high-quality spherical metal powers relies mainly on the atomization and rotating electrode methods, which need high-purity raw materials [12]. The development of the radio frequency plasma (RF) technique provides a powerful tool to reduce the costs of spherical metal powders [13]. This approach has been intensively researched for the manufacturing of cost-effective titanium spherical powders using low-cost hydrogenated–dehydrogenated (HDH) titanium powder [14]. Combining the advantages of the manufacturing capacity of AM technology and spherical HDH titanium powder with RF treatment, in this study, AM-HDH-RF-produced Ti components achieved superior mechanical properties and were prepared at a low-cost compared to those prepared using traditional processing techniques [15]. The HDH titanium powder used in this work was made of sponge titanium, machining residues, etc., as raw materials, and could be used for laser deposition after the radio frequency plasma (RF) process. Comparing with AM using titanium wire, this method omits the process of suspension melting, or arc melting, and the process of preparing wire, which has great advantages in terms of improvements to energy conservation and efficiency [16,17,18].

## 2. Experimental Section

The experimental apparatus for the HDH titanium powder treatment consisted of a radio frequency (RF), an inductively coupled plasma torch, a water-cooled steel chamber, a powder feeder, and a vacuum system. The plasma torch had a two-part structure, a water-cooled three-layer quartz confinement tube and a four-turn copper coil with water cooling. The plasma operated at an oscillating frequency of 3.5 MHz and used a 40 kW plasma plate. The central plasma and sheath gas used high-purity argon at flow rates of 20~30 L·min^−1^ and 80~90 L·min^−1^, respectively. The HDH-Ti powder was supplied by Jiangxi Weila Metal Materials Co., Ltd., Ganzhou, Jiangxi, China and had an oxygen content of 723.87 ppm and a nitrogen content of 9.38 ppm. The purity of the HDH-Ti powder was over 99.9 wt%. The average size of the HDH powder was about 135 μm, and was measured using a particle size analyzer (model Microtrac-S3500). The HDH-Ti powder entered the plasma arc via the carrier gas (argon). The flow rates of the carrier gas were 30 g/min, 45 g/min, and 60 g/min. Powder particles rapidly absorbed heat and spheroidized in the plasma arc, and finally entered the cooling chamber to rapidly condense and form spherical powder. 

Laser directed energy deposition (LDED) has the characteristics of net-forming and rapid-forming, and is one of the most promising AM methods in the preparation of complex metal components. In this work, the AM process was performed using LDED in an LDM6050 (YuChen Tech. Ltd., Nanjing, Jiangsu, China). The HDH titanium powder with RF treatment was processed in a drying oven at 393 k for 4 h under vacuum conditions. Specimens were produced on a pure Ti substrate under an argon atmosphere with an oxygen content less than 10 ppm. The parameters of the optimized process were as follows: a laser beam wavelength of 1070 nm, laser beam spot size of ~2 mm, laser power of 800 W, laser scanning speed of 600 mm/min, powder feeding rate of 9~14 g/min, coaxial argon gas flow of 12~20 L/min, and a sheath gas flow rate of 6~10 L/min. Figure 1 shows a schematic of the RF technique and the LDED for fabricating the CP-Ti sample. The cast Ti samples were fabricated via arc melting in a copper mold under a high-purity argon atmosphere. The raw material was at least 99.9 wt%.

Metallographic characterizations were performed using an ECLIPSE LV 150N optical microscope (OM, NIKON CORPORATION, Shinogawa Intercity Tower C, Konan, Minato-ku, Tokyo, Japan). The metallographic samples were mechanically polished and then etched in a solution of 10% HF, 5% HNO_3_, and 85% deionized water. The microstructures of the metallographic samples and the powder samples were characterized using scanning electron microscopy (SEM, LEO1530, LEO Electron Microscopy Ltd., Krefeld, Germany). A Microtrac-S3500 laser particle size analyzer was used to measure the powder size. Phase types were investigated using a Rigaku X-ray diffractometer(XRD, D/max-RB, Rigaku, Japan) using Cu-Kα radiation. The tensile properties of samples were examined at an ambient temperature using an Instron 5982 static testing machine with a strain rate of 10^−3^ s^−1^. To confirm that the results were reliable, three tensile samples from each group were tested.

## 3. Results and Discussion

The morphology of the HDH titanium powders, before and after induced RF treatment, were studied using SEM and the granularity of the powders were measured with a laser granularity analyzer. Figure 2 shows the microstructures of HDH titanium powders, before and after induced RF treatment. It can be seen in Figure 2a that the HDH titanium powder precursor has a rough surface in terms of morphology. There are some scraps and cracks located on the surface of the powders. For the HDH titanium powders with an RF treatment with a powder feeding rate of 30 g/min (Figure 2b), most of the irregular powders were changed into the spherical powders. However, with an increase of powder feeding rate from 30 g/min to 60 g/min, the proportion of non-spherical powders increased, as shown in Figure 2c,d. Based on the above results, it was revealed that RF treatment can markedly improve the morphology of HDH titanium powders. When powders are injected into a high-temperature plasma region, they rapidly absorb energy, resulting in the surfaces of the powders being heated to melt and are then subsequently quenched into a spherical shape. In the process of RF treatment, a lower powder feeding rate benefits the contact between powders and plasma. Therefore, the production efficiency of spherical titanium powders will be reduced with an increase in the powder feeding rate.

The particle size distributions of the HDH titanium powders, before and after RF treatment, are shown in Figure 3. Compared to powders without RF treatment, the peaks of the particle size distributions after RF treatment showed a lower position, indicating that the RF treatment has the ability to refine the HDH titanium powders. Furthermore, the peaks of the particle size distributions moved to a lower position with a reduction in the powder feeding rate. Some powders with a loose structure that are fragile cannot be subjected to the pressure caused by the thermal expansion and the residual titanium hydride dehydrogenation reaction, leading to the powders being broken up into a finer powder [19]. Therefore, there was more fine powder with the lower powder feeding rate.

Figure 4 shows the XRD spectrum of HDH titanium powders, before and after RF treatment. Peaks that represent the close-packed hexagonal (HCP) Ti crystal structure were observed in the XRD spectra of the original powder without RF treatment. At the same time, weak peaks belonging to the Ti_2_H phase were also observed in the enlargement of the XRD spectra of the original powder, as shown in Figure 4b, revealing that the process of dehydrogenated of the HDH Ti powder was incomplete. For the HDH titanium powders with RF treatment, the peaks showed that the Ti_2_H phases faded away in the XRD spectrum along with the reduction in the powder feeding rate. The above results indicated that the radio frequency plasma treatment, not only optimized the shape of the titanium powder, but also benefitted the removal of the residual Ti_2_H phase in the powder.

In order to analyze the effect of HDH powder and HDH-RF powder on the LDED-produced CP-Ti wall, the microstructures of the samples were observed with OM. The OM images in Figure 5a show that the morphology of the cast sample exhibited a specific dendritic grain microstructure. In Figure 5b, many blowhole defects with diameters of ~300 μm were left on the LDED samples prepared using HDH powder. Figure 5c shows an OM image of the LMD samples prepared using HDH-RF powder with a feeding rate of 30 g/min, which presents an almost-dense microstructure without porosity, revealing that HDH-RF powder is more suitable for the LDED technique than HDH powder. Furthermore, Figure 5d shows an enlarge image of the square in the Figure 5c, presenting that the microstructures of the LDED samples prepared using HDH-RF powder were composed of a mix of coarse and lath-shaped grains. Diverse microstructural features between the cast and LDED samples could be explained by the heating history, where significantly higher cooling rates were used in the process of LDED, leading to martensitic β transformation in CP-Ti solidification to an α phase with lath-shaped grains [20].

The tensile stress–strain curves of the titanium samples are shown in Figure 6. Compared with the LDED-HDH-produced samples and the cast samples, the LDED-HDH-RF-produced samples possessed an excellent combination of tensile strength and tensile ductility. It is well-known that defects in the samples cause performance degradation. The blowhole defects in the LDED-HDH-produced samples resulted in lesser mechanical properties compared to those of the LDED-HDH-RF-produced samples with an almost-dense microstructure. A fine microstructure results in a high yield strength, which is determined according to the Hall–Petch relation law, providing an increase in the yield stress of a polycrystalline material as its interplanar spacing decreases [21]. For CP-Ti, the yield strength is inversely proportional to the α grain size. The fine α lath shape obtained in LDED-HDH-RF-produced samples may be related to the high cooling rate during the LDED process, which promoted the β phase to α phase, with a fine net microstructure transformation [15]. Therefore, LDED-HDH-RF-produced samples with a lath shape showed a higher yield strength than the cast samples with equiaxed grains. The improved yield strength by employing the LDED-HDH-RF technique can enhance an implant CP-Ti material’s resistance to permanent shape change, which may increase its effectiveness for implant applications.

## 4. Conclusions

Hydrogenated–dehydrogenated (HDH) titanium powder has been employed to produce a spherical Ti powder using the RF technique. The spherical Ti powder was also used as the raw materials for LDED to create the CP-Ti. Spherical Ti and CP-Ti samples were further analyzed by means of XRD and SEM. The main conclusions were described as follows:(1)Microstructural analyses of the powder revealed that the radio frequency plasma treatment, not only optimizes the shape of titanium powder, but also benefits the removal of the residual Ti_2_H phase in the powder.(2)Microstructural analyses of the CP-Ti samples present that the LDED-HDH-RF-produced samples have a denser microstructure than the LDED-HDH-RF-produced samples.(3)Tensile testing indicated that LDED-HDH-RF-produced samples possess an excellent combination of tensile strength and tensile ductility. Compared to the cast samples, the higher yield strength of the LDED-HDRHT-produced samples should be attribute to the fine α lath shape.(4)The present work provides an approach for LDED-fabricated CP-Ti to accommodate the economic and mechanical properties of CP-Ti, while providing insights into the expanding application of HDH titanium powder.

## Figures and Tables

**Figure 1 materials-15-01548-f001:**
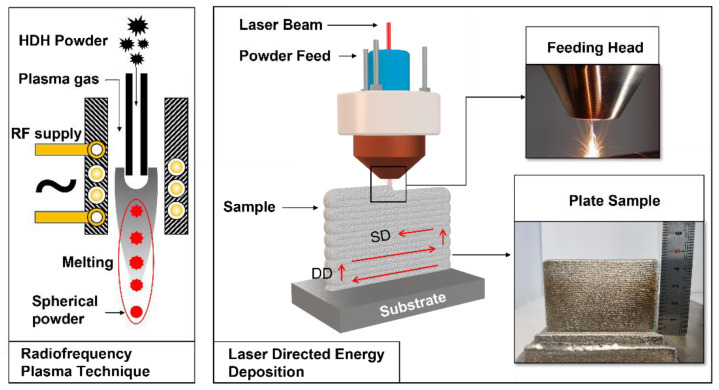
Schematic of the RF technique and the LDED fabrication process for Ti samples.

**Figure 2 materials-15-01548-f002:**
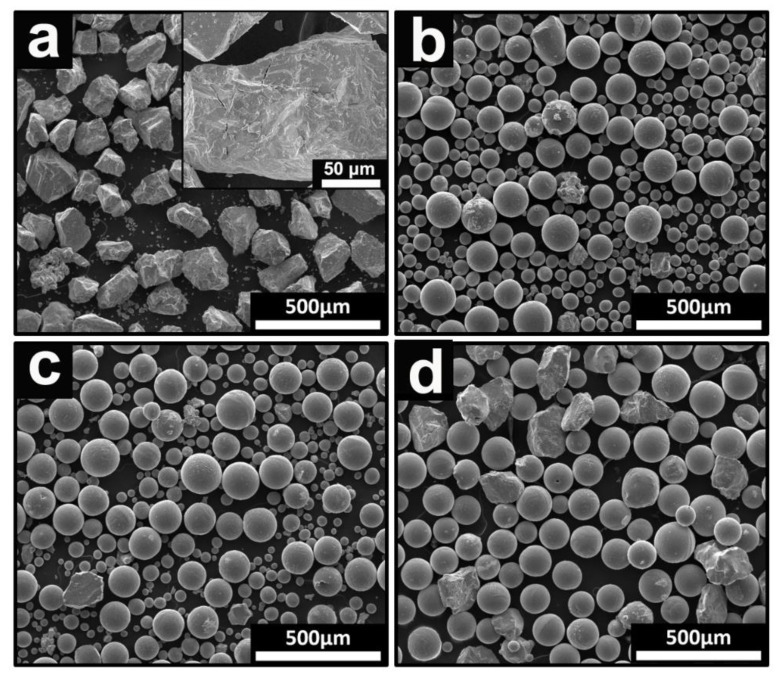
The micrograph show HDH titanium powder under different parameters: (**a**) without RF, (**b**) RF with a feeding rate of 30 g/min, (**c**) RF with a feeding rate of 45 g/min and (**d**) RF with a feeding rate of 60 g/min.

**Figure 3 materials-15-01548-f003:**
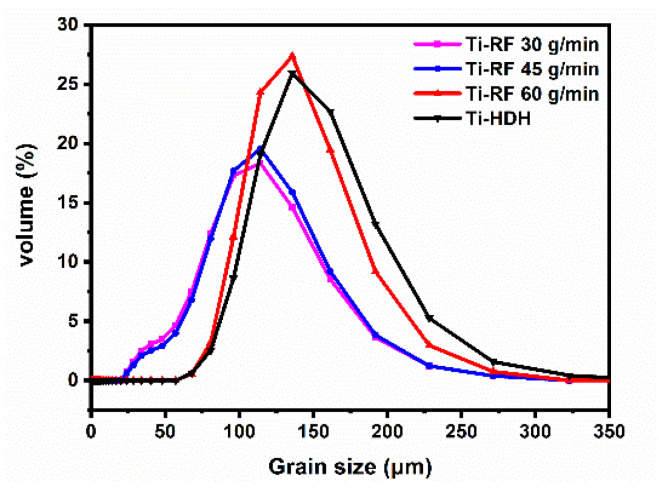
The particle size distributions of HDH titanium powders, before and after radio-frequency plasma treatment.

**Figure 4 materials-15-01548-f004:**
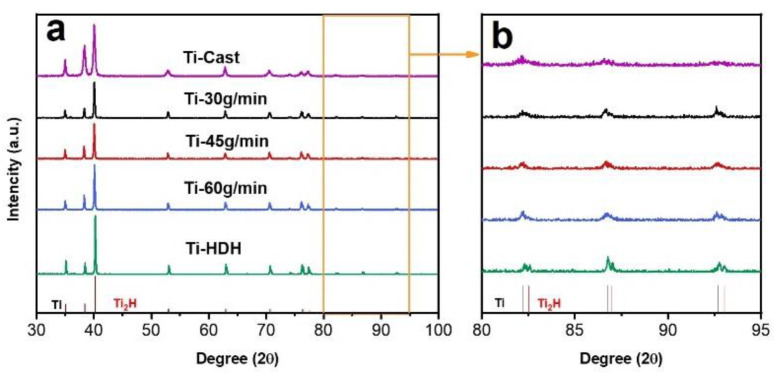
(**a**) XRD pattern of HDH titanium powders before and after radio frequency plasma treatment; (**b**) zoom-in image of the XRD patterns with 2θ ranging from 80° to 95°.

**Figure 5 materials-15-01548-f005:**
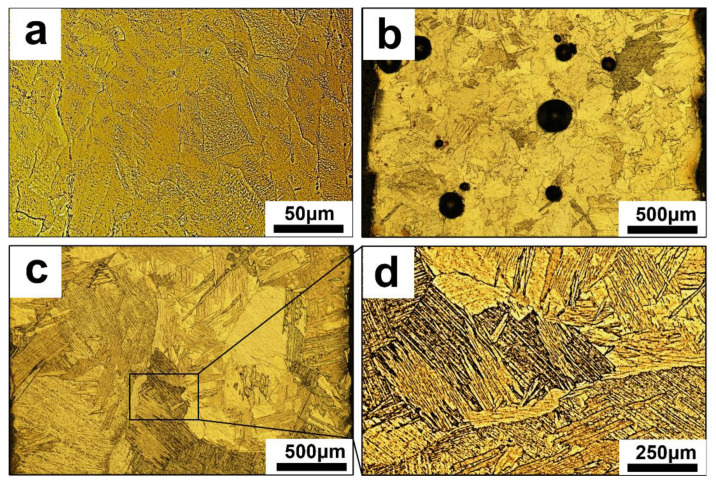
The optical microscope of CP-Ti using different prepared methods: (**a**) cast, (**b**) LDED-HDH titanium, (**c**) LDED-HDH-RF titanium, (**d**) zoom-in image of the square in LDED-HDH-RF titanium.

**Figure 6 materials-15-01548-f006:**
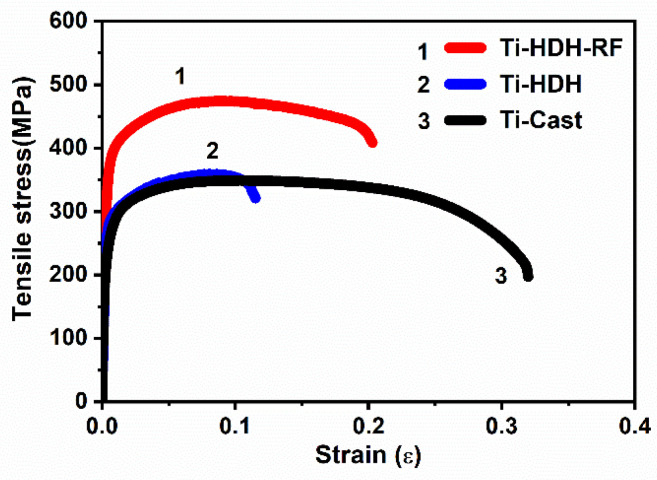
The tensile stress–strain curves of CP-Ti produced using different prepared methods.

## Data Availability

The data used to support the findings of this study are available from the corresponding author upon request.

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
