# Peer review of "Effect of Radiofrequency Plasma Spheroidization Treatment on the Laser Directed Energy Deposited Properties of Low-Cost Hydrogenated-Dehydrogenated Titanium Powder"

_materials, 2022, doi:10.3390/ma15041548_

Round 1

Reviewer 1 Report

The paper “Effect of radiofrequency plasma spheroidization treatment on the laser melting deposited properties of low-cost Hydrogenated-dehydrogenated titanium powder” is well-organized. The work is complete, with different kind of analyses.

The authors studied the effects of the radiofrequency plasma (RF) technique to produce the spherical CP-Ti using powder low cost hydrogenated-dehydrogenated (HDH) titanium powder in conjunction with laser metal deposition. The evolution of the CP-Ti microstructure and mechanical properties response accommodate the economy and mechanical properties of the CP-Ti.

In my opinion, after problems have been addressed, I suggest to publish it onto Materials after minor revisions.

  1. In Experimental the authors should provide a specific paragraph about technical data and the supplier on the commercial HDH titanium powder used.
  2. Also provide the type of optical microscope and the chemical etch used to investigate the microstructure.
  3. I think it would be better to add some information about where the tensile stress-strain curves of CP-Ti were ploted.
  4. Keywords should be arranged alphabetically. They are partially numbered.
  5. Figures 1, 2 and 5 should be larger because it is difficult to see relevant details.
  6. Figure 2a provide in insert a higher magnification SEM. In my opinion it would be better (for comparison) to add higher magnification on RF-HDH titanium powder used in Figure 5c and d.
  7. Powder produced via RF with feeding rate 60 g/min (Fig.2d) seams to presents about the same size as sample produce by RF with feeding rate 45 g/min. This is not correlate with data in figure 3. Also it can be observed a lot powder with a rough surface.
  8. Which is the RF feeding rate for the sample from the figure 5c and d? Please mention it.
  9. The same question for RF-HDH powder analyzed in Fig 6 and Table 1.
  10. Lines 151 and 152 “The OM images in Fig.5(a) shows the morphology of cast sample is filled by equiaxed grains.” In as-cast state the specimens exhibit specific dendritic grain microstructure, respectively chemically inhomogeneous solid solution. Image resolution is very low and it is very difficult to observe the grains boundary of equiaxed structure. The authors should improve the quality of this image or to highlight one grain boundary or provide optical micrograph at the same magnification as images b and c.
  11. Please correct the Fig. 5 caption, since it can be see that the image from Figure 5d is not OM.

Reviewer 2 Report

  1. Figure 1 is not clear. Please replace it. The wording is too small. Please comment on why you presented this figure. It is a well-known concept for DED systems.
  2. Add a short note about the results to the abstract.
  3. How authors selected the process parameters for the experimentation. This is important for repeating the test.
  4. The text has some typos. Please check them.
  5. The introduction needs to be updated by comparing the DED of the wire feed with the following references.
  • Tang, S., G. Wang, H. Song, R. Li, and H. Zhang, A novel method of bead modeling and control for wire and arc additive manufacturing.
  • Fang, X., C. Ren, L. Zhang, C. Wang, K. Huang, and B. Lu, A model of bead size based on the dynamic response of CMT-based wire and arc additive manufacturing process parameters.
  • Kulkarni, J.D., S.B. Goka, P.K. Parchuri, H. Yamamoto, K. Ito, and S. Simhambhatla, Microstructure evolution along build direction for thin-wall components fabricated with wire-direct energy deposition.
  1. Additive manufacturing has many advantages over the conventional manufacturing method which can be highlighted in your paper. Please read the following article and add to the introduction to show the experimental application of additive manufacturing and the advantage of this process over conventional manufacturing like machining.

Additive manufacturing a powerful tool for the aerospace industry.

  1. Please add a comment on the repeatability of your testing and results.

Reviewer 3 Report

The manuscript by Min Liu et al. entitled: “Effect of radiofrequency plasma spheroidization treatment on the laser melting deposited properties of low-cost Hydrogenated-dehydrogenated titanium powder” presents an experimental work on the production of commercial pure titanium samples by laser directed energy deposition technique. The work discuss the properties of the samples produced from different types of powders. Here are the main aspects that need to be amended/corrected:

  1. During the development of laser-assisted additive manufacturing methods, there have been numerous different terms used by different authors that refer to basically the same techniques. In order to avoid further confusion in the readers, it is envisaged to follow the existing standards that stablish the terminology to be used to name the different additive manufacturing techniques. Therefore, in the present manuscript, instead of using Laser Metal Deposition (LMD), please use Laser Directed Energy Deposition (LDED) as stated in the ISO/ASTM 52900:2015 international standard.

  1. Please update the literature review including recent works on LDED of pure titanium as the following ones:

Attar, H. et al.Evaluation of the mechanical and wear properties of titanium produced by three different additive manufacturing methods for biomedical application. Mater. Sci. Eng. A 2019, 760, 339–345.

Amado, J.M. et al. A comparison of laser deposition of commercially pure titanium using gas atomized or Ti sponge powders. Surf. Coatings Technol. 2019, 374, 253–263.

Barro, Ó. et al. Improved Commercially Pure Titanium Obtained by Laser Directed Energy Deposition for Dental Prosthetic Applications. Metals 2021, 11, 70.

  1. Please give details about the type of laser used, wavelength, beam quality, beam diameter on the substrate, focusing lens, etc … .

  1. Even if the starting material is 99.9 pure titanium, a very small amount of Fe or other trace elements can induce intergranular segregation that can influence on the mechanical properties of the additively manufactured part. Please give a complete analysis of the powders and the produced samples.

  1. The reported values of the tensile strength for the laser produced samples are lower than those reported in recent works. Please discuss in the manuscript the possible reasons for this fact.

Round 2

Reviewer 2 Report

The paper is ready for publishing.

Reviewer 3 Report

Authros corrected/amended the manuscript according to suggestions. The manuscript can be now accepted for publication.